# Molecular Pathways Involved in the Development of Congenital Erythrocytosis

**DOI:** 10.3390/genes12081150

**Published:** 2021-07-28

**Authors:** Jana Tomc, Nataša Debeljak

**Affiliations:** Medical Centre for Molecular Biology, Institute of Biochemistry and Molecular Genetics, Faculty of Medicine, University of Ljubljana, SI-1000 Ljubljana, Slovenia; jana.tomc@mf.uni-lj.si

**Keywords:** congenital, familial, erythrocytosis, disease mechanisms, signal transduction, transcriptomic, proteomic, metabolomics

## Abstract

Patients with idiopathic erythrocytosis are directed to targeted genetic testing including nine genes involved in oxygen sensing pathway in kidneys, erythropoietin signal transduction in pre-erythrocytes and hemoglobin-oxygen affinity regulation in mature erythrocytes. However, in more than 60% of cases the genetic cause remains undiagnosed, suggesting that other genes and mechanisms must be involved in the disease development. This review aims to explore additional molecular mechanisms in recognized erythrocytosis pathways and propose new pathways associated with this rare hematological disorder. For this purpose, a comprehensive review of the literature was performed and different in silico tools were used. We identified genes involved in several mechanisms and molecular pathways, including mRNA transcriptional regulation, post-translational modifications, membrane transport, regulation of signal transduction, glucose metabolism and iron homeostasis, which have the potential to influence the main erythrocytosis-associated pathways. We provide valuable theoretical information for deeper insight into possible mechanisms of disease development. This information can be also helpful to improve the current diagnostic solutions for patients with idiopathic erythrocytosis.

## 1. Introduction

Oxygen homeostasis involves several tissues and organs, including the heart, lungs, bone marrow and blood. Any imbalance in its homeostatic mechanisms can result in hematological disorders, such as anaemia or erythrocytosis. Most often erythrocytosis is the compensatory mechanism of heart or lung disease (secondary acquired erythrocytosis). Rarely erythrocytosis is a consequence of single gene variant (monogenic disorder), that can be acquired (*JAK2*, indicative for polycythaemia vera, PV) or inherited (*EPOR, VHL, EGLN1, EPAS1, EPO, HBB, HBA, BPGM*, indicative for congenital erythrocytosis, ECYT) [1,2]. The genetic disorder can also be caused by the combined action of more than one gene (digenic or polygenic disorder). However, this mechanism is not well addressed in the case of erythrocytosis. An update on the currently known causes of inherited erythrocytosis can be reviewed in publication within this issue by M.F. McMullin (2021) [3].

After excluding the mutation in *JAK2*, patients with persistent erythrocytosis from a young age or with a family history are usually screened for mutations in 9 genes involved in oxygen sensing in kidneys (*EPOR*, *VHL*, *EGLN1*, *EPAS1*, *EPO*), erythropoietin signal transduction in pre-erythrocytes (*EPOR*) and hemoglobin-oxygen affinity regulation in mature erythrocytes (*HBB, HBA, BPGM*). However, in more than 60% the genetic cause remains undiagnosed, suggesting that other genes and mechanisms must be involved in the disease development. Understanding the main regulatory pathways in the process of oxygen homeostasis is necessary to identify new potential factors, which have not been yet associated with erythrocytosis but could contribute to its development. Besides, this can further improve the diagnostic methods for prompt and accurate diagnosis and selection of appropriate treatment.

In this paper, we aim to review molecular mechanisms correlated with known oxygen sensing HIF-EPO pathway, EPO-EPOR signal transduction pathway, and hemoglobin-oxygen (Hb-O_2_) affinity modulation. Furthermore, additional new mechanisms that could be implicated in the development of erythrocytosis are investigated, including regulation of erythropoiesis, metabolism, and hormone homeostasis. Comprehensive research of the literature was performed and different in silico tools were used, including Reactome [4], String [5], UniProt [6], GeneCards [7], Human Protein Atlas [8], NCBI [9], and HGNC [10] databases. We carefully revised the mechanisms involved in regulation of RNA transcription, as the expression level of key players can extensively moderate described pathways. Furthermore, regulation at the protein level was studied in details, identifying mechanisms that affect protein-protein interactions, protein modifications, and membrane transport. Regulation of signal transduction is important for all extracellular signals, as signaling pathways can be attenuated or enhanced. Non the less, the revision of metabolism is necessary, as modulation of glucose and iron metabolism was confirmed to be involved in the development of erythrocytosis [11,12].

## 2. Known Molecular Pathways Involved in Congenital Erythrocytosis

### 2.1. Oxygen-Sensing: HIF-EPO Pathway

Often named also the HIF-EPO pathway after its key molecules, this signaling cascade has a prominent role in oxygen sensing by hypoxia-inducible factors (HIF). In tissue hypoxia, the transcription factor HIF-α will dimerize with HIF-β and bind to hypoxia-response elements (HRE) on promotor of several tissue-specific proteins, including erythropoietin (EPO). Accumulation of HIF2α, the main HIF-α subunit in kidneys, will result in EPO synthesis and increased erythrocyte production in the bone marrow. In normoxia, HIF2α is degraded by prolyl hydroxylase (PHD2) and von Hippel-Lindau (VHL) protein, while in hypoxia PHD2 is not active resulting in HIF2α accumulation.

Any imbalance in molecular mechanisms of HIF-EPO pathway can result in hematological disorders. Four types of erythrocytosis are known to be caused by mutations in genes encoding key players of oxygen-sensing signaling, namely *VHL* (ECYT2), *EGLN1*/PHD2 (ECYT3), *EPAS1*/HIF2α (ECYT4) and *EPO* (ECYT5) [13]. These mutations mainly result in HIF2α accumulation in normoxia, due to loss of VHL and PHD2 function (impaired degradation) or gain of HIF2α function (increased stability). The role of *EPAS1* paralogues (HIF-α subunits HIF1α and HIF3α), *ARNT* (HIF-β subunit HIF1β) and *EGLN1* paralogues (*EGLN2*/PHD1 and *EGLN3*/PHD3) in development of erythrocytosis has also been studied, but it has not been confirmed. Targeted exome NGS analysis of patients with idiopathic erythrocytosis by Camps et al. (2016) [14], including known disease-causing oxygen-sensing genes (*EPAS1*, *EGLN1*, *VHL*) and novel candidate genes from oxygen-sensing pathway (*EPO*, *HIF1A*, *HIF3A*, *HIF1AN*, *EGLN2*, *EGLN3*, *BHLHE41*, *OS9*, *ZNF197*, *KDM6A*) confirmed variants in all known disease-causing genes and identified five novel candidate genes associated with ECYT. The novel gene variants found within this study were located in *EPO*, *EPAS1* paralogue *HIF3A*, *EGLN1* paralogue *EGLN2,* and two factors associated with HIF; OS9 Endoplasmic Reticulum Lectin (*OS9*) and class E basic helix-loop-helix protein 41 (*BHLHE41* or SHARP1) [14]. Furthermore, the role of DNA non-coding regions in development of erythrocytosis should further be considered, as it has already been confirmed in *EPO* and *VHL* by functional studies [15,16].

During our in silico research, we identified new candidate genes involved in transcriptional activation or repression, post-translational modifications, and nuclear transport. These mechanisms can directly or indirectly modulate the stability or activity of key transcription factor HIF2α and affect the process of EPO production (for a recent review see [17]). However, as HIF2α also plays a role in cancer development and progression and the greater number of available information is cancer-directed, their enrolment in the regulatory mechanisms in healthy tissue must be considered with caution. The key players of the oxygen-sensing pathway are presented in Figure 1. Regulation of *EPAS1* is extensively revised as an integrative regulatory map within this special issue.

#### 2.1.1. Regulation at the RNA Level

Oxygen sensing depends on the tight regulation of all key players, including *VHL*, *EGLN1*, *EPAS1* and *ARNT*.

The HIF1β subunit (encoded by gene *ARNT*) is constitutively expressed in all tissues, while the expression of three HIFα subunits is tissue specific. There is not much information available about the regulation of HIF2α (encoded by gene *EPAS1*) expression in the kidney and liver. Depending on the cell line used, HIF2α mRNA is regulated by transcription factor E2F1, deubiquitylase Cezanne, transcription factors SP1 and SP3 and others [19,20]. Apart from that, MCP-induced protein 1 and DNA methyltransferases (DNMTs) have been reported to negatively regulate the transcription of *EPAS1* gene [21,22]. HIF also regulated the RNA expression of transferrin (TF) and transferrin receptor (TFR), the key players in iron metabolism (see Section 3.2).

PHD2 (encoded by gene *EGLN1*) promoter contains an HRE region and its transcription is positively regulated by HIF1α [23,24]. Besides, transforming growth factor β-1 (TGFβ1) has been demonstrated to negatively regulate PHD2 expression through SMAD signaling in HepG2 cells [25].

The analysis of specific transcription factor binding sites within the *VHL* promoter identified a significant positive regulatory element as an SP1 binding site [26]. Furthermore, FOXO3a has been recently reported to play a role in hypoxia signaling by direct binding to the *VHL* promoter and upregulation of *VHL* expression [27]. HIF1α-dependent upregulation of *VHL* expression as a negative feedback loop during hypoxia has also been suggested [28].

*EPO* gene expression is regulated by HIF2α -HIF1β complex and p300/CBP transcriptional co-activators in cooperation with other transcription factors, such as hepatocyte nuclear factor-4α (HNF4α) [29], retinoid X receptor α (RXRα) [30], or nuclear receptor coactivator 3 (NCoA-3) [31]. Transcription factors GATA2 and GATA3 have been shown as negative regulators of *EPO* transcription [32,33]. Besides, methylation of CpGs in the promoter or 5′-untranslated regions represses *EPO* transcription by blocking the binding of sequence-specific DNA binding proteins or recruitment of methyl-CpG binding protein 2 [34]. An extensive study of one family with idiopathic erythrocytosis indicated, that transcription of *EPO* mRNA from an alternative promoter, due to the frameshift mutation in exon 2 that interrupts translation of the main *EPO* mRNA, can lead to excess production of erythropoietin and erythrocytosis [15]. *EPO* expression may be induced by several metals (cobalt, nickel and manganese) and by iron chelation (see Section 3.3).

#### 2.1.2. Regulation at the Protein Level

##### Protein Interactions and Modifications

HIFα proteins can be directly or indirectly modulated by post-translational modifications. Beside to O_2_ dependent hydroxylation, also phosphorylation [35,36,37], acetylation [38,39], methylation [40], ubiquitination [41] and sumoylation [42,43] have been found to affect its transcriptional activity or stability. However, many of these post-translational mechanisms do not require O_2_ as a co-factor.

Apart from post-translational modifications, other protein interactions can regulate the activity of HIFs. Several modulators of HIF2α are known, such as NF-kappa-B essential modulator (NEMO) and class E basic helix-loop-helix protein 41 (bHLHe41) [44,45]. Likewise, many proteins interact with PHD2 and VHL and consequently regulate HIF2α hydroxylation or ubiquitination and HIF2α stability [46,47,48,49,50,51]. VHL is the substrate recognition component of Cullin Ring Ubiquitin ligase complex (CRL), consisting of elongins B and C (ELOB and ELOC), scaffold protein cullin-2, and ubiquitin-protein ligase RBX1, and all these components play an important role in VHL-dependent ubiquitination and degradation of HIF2α [52]. Besides, transforming acidic coiled-coil-containing protein 3 (TACC3) can act as a co-activator of the HIF complex by interacting with constitutively expressed HIF1β subunit [53].

##### Nuclear Transport

Intracellular transport of the oxygen-sensing machinery represents an important level of control of transcriptional activity of HIF subunits in oxygen homeostasis. Unlike HIF2α, which shuttles between nucleus and cytoplasm, HIF1β is permanently inside the nucleus [54]. Importins α1, α3, α5 and α7 bind to both HIF2α and HIF1β, allowing their translocation into the nucleus. HIF2α can also associate directly with importin β which mediates the interaction with the nuclear pore complex [55]. Nuclear export of HIFα subunits is regulated by exportin 1 (CRM1), and HIF2α has been sown to associate with CRM1 in a phosphorylation-sensitive manner regulated by MAP1/3 kinases [56]. Transport between nucleus and cytoplasm has been reported also for PHD2 and VHL, wherein nuclear export of the two proteins involves CRM1 [54] or elongation factor 1-α 1 (EEF1A1) [57].

### 2.2. Signal Transduction: EPO-EPOR Pathway

EPO stimulates proliferation, survival and differentiation of erythroid progenitors in the bone marrow by binding to EPO receptors (EPOR) expressed on their surface. Tyrosine phosphorylation plays a pivotal role in EPOR signal transduction and the activation of downstream signaling cascades involved in erythropoiesis, including the signal transducer and activator of transcription 5 (STAT5), phosphatidylinositol 3-kinase (PI3K) and mitogen-activated protein kinase (MAPK/ERK) pathways [58,59,60]. The key players of EPO-EPOR signal transduction have been recently reviewed and are presented in Figure 2 [61,62].

Abnormally increased or prolonged activation of the three pathways is implicated in deregulated erythropoiesis and overproduction of red blood cells in polycythemia vera and congenital erythrocytosis [59], wherein inherited mutations in the *EPOR* itself (ECYT1), and acquired mutations in *JAK2* (PV) are already well known genetic causes [1,63]. All mutations result in prolonged EPOR signal transduction, due to a gain of JAK2 and EPOR function. The *JAK2* mutation results in the impaired auto-inhibitory domain, and consequently prolonged JAK2 autophosphorylation and STAT5 pathway activation, while some *EPOR* mutations result in the loss of EPOR negative regulatory mechanisms, such as loss of SHP1 binding site, and consequently prolonged JAK2 autophosphorylation and STAT5 pathway activation [64]. Occasionally, mutations in gene encoding SH2B adapter protein 3 (LNK; *SH2B3*) may account for or at least contribute to the erythrocytosis phenotype, and it has already been suggested that SH2B3 mutations should be a part of the investigation of idiopathic erythrocytosis [65,66]. Furthermore, severe erythrocytosis has been observed in *SOCS3* null mice [67]. Targeted exome NGS analysis of patients with idiopathic erythrocytosis by Camps et al. (2016) included three known disease-causing genes involved in EPOR signal transduction (*EPOR, SH2B3, JAK2*) and confirmed their role in the development of erythrocytosis [14]. Rare *JAK2* germline mutations have been associated with the hereditary type of erythrocytosis [68]. Several somatic mutations in the genes *JAK2*, *TET2, NFE2, KMT2A* and/or *TP53* have been found in cases of polycythemia vera [69].

During our in silico research we identified new candidate genes encoding adaptor proteins, tyrosine kinases and phosphatases that participate in phosphorylation and activation, and especially in negative regulation of EPOR and the regulation of STAT5, PI3K and MAPK signaling cascades. Besides, genes involved in nuclear transport and regulation of EPOR mRNA expression should not be neglected. The contribution of these mechanisms to abnormal signal transduction and deregulated erythropoiesis should be further investigated. Mathematical modeling may facilitate future research, as a similar model of the JAK2/STAT5 signaling pathway, used in cancer research, can be applied also for modeling erythrocytosis [70].

#### 2.2.1. Regulation at the RNA Level

Erythroid transcription factor GATA1 plays a central role in erythroid development, and its binding motifs have been identified in *EPOR* promoter/enhancer. Signaling through EPOR can increase the expression and activation of GATA1 which in turn upregulates *EPOR* mRNA expression in late CFU-E/proerythroblast progenitors [64]. However, the mRNA expression of *EPOR* is tightly regulated by GATA1 in coordination with transcription factors SP1 and TAL1, which both stimulate the transcription [71,72,73]. Especially SP1 has been shown as a critical element in transcriptional activation of the *EPOR* promoter [72].

Transcriptional regulation of key downstream players of EPOR signaling cascade shall also be revised, including JAK2, STAT5, PI3K, and MAPK. Furthermore, the interplay between epigenetics and transcriptional regulation must be further addressed [74,75].

#### 2.2.2. Regulation at the Protein Level

##### Protein Interactions and Modifications in STAT5 Pathway

STAT5 refers to two proteins, which share 94% structural homology and are transcribed from *STAT5A* and *STAT5B* genes [76]. Both proteins are predominant signal transducers for EPOR and essential in regulating erythroid differentiation and survival [76,77].

Activation of this pathway involves STAT5 recruitment to the phosphorylated EPOR and tyrosine phosphorylation by JAK2, dissociation from the receptor, and formation of homodimers [78]. STAT5 dimers are recognized by importins α3 and β, allowing them to translocate into the nucleus [79]. Together with other transcription factors, such as TAL1, GATA1, and KLF1, STAT5 dimers associate with promoter and enhancer regions of target genes to upregulate their expression [75]. Recently, more than 300 STAT5-occupied sites in response to EPO have been identified, including many new genes involved in driving erythroid cell differentiation [80].

Besides JAK2, several members of the Src family tyrosine kinases have been reported to phosphorylate tyrosine residues of STAT5, such as proto-oncogene tyrosine-protein kinase SRC [81], tyrosine-protein kinase FYN [78], and tyrosine-protein kinase LYN [58]. LYN is also involved in the tyrosine phosphorylation of EPOR itself, as well as activates other signaling molecules [82]. Another signaling molecule that associates with and modulates STAT5 activity in response to EPO is CRK-like adapter protein (CRKL) [83]. Furthermore, 22 new EPO-modified kinases and phosphatases have been recognized in a study by Held et al. (2020) [84], including protein tyrosine phosphatase non-receptor type 18 (PTPN18), which is highly expressed in hematopoietic progenitors. PTPN18 has been shown not only to promote STAT5 signal transduction for EPO-dependent hematopoietic cell growth, but also signal transduction of PI3K and MAPK pathways [84].

##### STAT5 Negative Regulation

STAT5 activity is inhibited through inhibitory mechanisms of protein-tyrosine phosphatase SHP1, SH2B adapter protein 2 (SH2B2) and 3 (SH2B3). SH2B3 (synonym LNK), for example, blocks the STAT5 pathway by inhibiting EPOR phosphorylation and JAK2 activation [78,85], and recruitment of SHP1 to a segment of the EPOR causes dephosphorylation of JAK2 and down-regulation of EPOR signals [86,87]. Suppressors of cytokine signaling (SOCS) 1, 2 and 3, and cytokine-inducible SH2-containing protein (CISH) have been shown as the most potent inhibitors of EPO signaling [67]. CISH associates with activated EPOR and recruits a complex of elongin B, elongin C, cullin5 and RBX2, to provide E3 ubiquitin ligase activity towards nearby substrates, including STAT5, leading to their degradation [80]. SOCS3 is especially essential to control the STAT5 phosphorylation levels. Apart from functioning in a similar way to CISH, SOCS3 also binds directly to JAK2 as well as to the activated EPOR and inhibits EPO-dependent cell proliferation and STAT5 activation [67]. SOCS3 activity might be inhibited in the absence of SOCS2, as SOCS2 knock-down in MDS cell lines resulted in hyper-phosphorylation of STAT5 [88]. Furthermore, serine/threonine-protein kinase pim-1 (PIM1) has been reported to interact with and potentiate inhibitory effects of SOCS1 and SOCS3 on STAT5 [89]. Genes encoding PIM1, CISH, and SOCS3 have been identified as STAT5 target genes. Another gene regulated by STAT5 and involved in negative regulation of EPOR signaling is clathrin interactor 1 (CLINT1), which contributes to receptor internalization [80].

##### Protein Interactions and Modifications in PI3K Pathway

PI3 kinase signaling plays an important role in EPO-induced proliferation and differentiation of erythroid progenitors, protecting the cells from apoptosis and regulating EPO-induced mitogenic responses. The central mediator of the PI3K pathway is serine/threonine kinase AKT (protein kinase B; PKB) [90], which regulates the activity or expression of important transcription factors and genes [91,92].

Phosphorylation of PI3K after EPO stimulation can be induced by binding of PI3K regulatory subunit (p85) to phosphorylated EPOR, or through mechanisms that involve GRB2-associated-binding proteins (GAB) 1 and 2 or insulin receptor substrate-2 (IRS2). All three mechanisms are equally active in primary erythroid progenitors [91,93]. Activated PI3 kinase further phosphorylates phosphatidylinositol 4,5-bisphosphate (PI-(4,5)-P2) to produce phosphatidylinositol 3,4,5-trisphosphate (PI-(3,4,5)-P3) which is an AKT activator. It recruits AKT to the cell membrane, causing AKT conformational alterations, and activates 3′phosphoinositide-dependent protein kinase 1 (PDK1) which phosphorylates AKT. AKT is double-phosphorylated by PKD2, separated from the membrane and translocated to the nucleus rapidly and transiently, where it further mediates its enzymatic effects [90,94].

##### PI3K/AKT Negative Regulation

The activity of AKT and its downstream signaling is negatively regulated by several phosphatases, such as phosphatase and tensin homologue (PTEN), and phosphatidylinositol-3,4,5-trisphosphate 5-phosphatases (SHIP) 1 and 2. While PTEN transforms the AKT activator PI-(3,4,5)-P3 into PI-(4,5)-P2, SHIPs dephosphorylate PI-(3,4,5)-P3 to PI-(3,4)-P2. The expression of SHIP1 is largely restricted to hematopoietic cells, while SHIP2 appears to be more widely expressed [90,95]. Another component of negative regulation of the AKT signaling is carboxyl-terminal modulator protein C (CTMP). CTMP has been shown to directly interact with AKT at the plasma membrane and to inhibit its phosphorylation [96].

Up to now, the performance of PI3K/AKT in the process of erythropoiesis has not been studied as much as other signaling pathways, and future research in this area could be beneficial for a better understanding of its regulatory mechanisms and contribution to the development of erythrocytosis.

##### Protein Interactions and Modifications in MAPK Pathway

One of the key pathways involved in erythropoiesis is the RAS/RAF/MAPK signal transduction cascade. Balanced activation of this pathway is important for promoting proliferation and anti-apoptosis of erythroid progenitors, and its downregulation has been indicated as critical for final erythroid maturation [97].

The pathway is activated by tyrosine phosphorylation of SHC-transforming protein 1 (SHC) that forms a complex with growth factor receptor-bound protein 2 (GRB2) and son of sevenless homolog 1 (SOS1) [87,98]. Besides, GRB2 can bind to tyrosine-phosphorylated CRKL protein, which is in hematopoietic cells recruited to phosphorylated EPOR after EPO stimulation and becomes tyrosine-phosphorylated by LYN [99]. EPO also induces tyrosine phosphorylation of SHIP, which has been shown to associate with EPOR and to form a ternary complex with SHC and GRB2, resulting in activation of MAPK pathway [100]. In EPO-treated UT-7 cells, tyrosine-phosphorylated IRS2 has been demonstrated to associate with SHIP [83]. Phosphatase SHP2 has also been reported to initiate the MAPK signaling pathway after EPO stimulation by recruitment to EPOR and GRB2 [101].

After interacting with GRB2, SOS1 is recruited to the plasma membrane where it activates GTPase RAS by stimulating the exchange of GDP for GTP. GTP-RAS recruits proto-oncogene serine/threonine-protein kinase RAF (RAF-1) to the cell membrane, thus enabling activation of RAF kinase [102,103]. Three human RAS proteins function as regulated GDP/GTP molecular switches, H-, K-, and N-RAS respectively, and they can all activate RAF-1 [104]. Active RAF-1 further phosphorylates and activates the dual specificity mitogen-activated protein kinase kinases (MEK) 1 and 2, and MEK1/2 phosphorylates the tyrosine and threonine residues of their only known physiological substrates mitogen-activated protein kinases (MAPK or ERK) 1 and 2 [105]. Activated ERK1/2 kinases phosphorylate a large number of substrates localized in the cytoplasm and nucleus, and for the latter, nuclear translocation of ERK1/2 is required and occurs through its interaction with importin 7 [87,104,106].

Other signaling molecules can additionally modulate individual components of the MAPK cascade. UCP2 protein, for example, can interact with ERK1/2 and regulate its phosphorylation in the process of erythropoiesis [107]. Also, RAF-1 activation can be increased through interaction with several kinases and other proteins, such as SRC kinase, protein kinase C (PKC) family members, serine/threonine-protein kinase (PAK) 3, JAK2 or kinase suppressor of RAS (KSR) [104,108]. Furthermore, malignant fibrous histiocytoma-amplified sequences with leucine-rich tandem repeats 1 (MASL1), a candidate oncogene, have been shown to interact with RAF-1 and activate erythroid differentiation [109].

##### MAPK Negative Regulation

ERK1/2 kinases are dephosphorylated and inactivated by serine/threonine or tyrosine phosphatases and dual-specificity MAP kinase phosphatases (MKPs), such as PP2A, PP2C, STEP, HePTP, PTP-SL, MKP-1, PAC1, hVH3, MKP-3, MKP-X [110]. MKPs activity, however, is also positively regulated by ERK1/2 kinases [94]. Recently, 6 novel phosphatases have been found to be implicated in the inhibition of the MAPK pathway, although their roles in erythropoiesis need further exploration [97]. RAS activity is negatively regulated through GTPase activating proteins (GAPs), such as RASA1, SynGAP or Neurofibromin, that promote the formation of inactive GDP-bound RAS [111]. EPO-induced MEK mediated hyper-phosphorylation of RAF-1 has been shown to result in inhibition of RAF-1 kinase activity towards MEK [98], and RAF kinase inhibitor protein (RKIP) has been demonstrated to disrupt the interaction between RAF-1 and MEK kinases, consequently inhibiting the phosphorylation and activation of MEK [112]. Spred proteins have been reported as general inhibitors of the MAPK signaling pathway, by binding to RAS and RAF-1 and suppressing the phosphorylation and activation of RAF-1 [113]. Importantly, Spred1 and Spred2 are induced by EPO in primary bone marrow-derived CFUe-like progenitors [114].

### 2.3. Hemoglobin-Oxygen Affinity Modulation

Hemoglobin (Hb) in mature erythrocytes is mainly responsible for the transport of O_2_ from the lungs to the tissues and CO_2_ from the tissues to the lungs. 98% of O_2_ in the bloodstream is in a Hb-bound state. To accommodate maximum space for Hb, erythrocytes lack the nucleus and most of the cytoplasmic organelles. This also allows the drastic modification of their biconcave shape, enabling the passage through microcapillaries and providing an optimal area for respiratory exchange [1,115]. The key players of hemoglobin oxygen affinity are presented in Figure 3.

Increased oxygen affinity of Hb causes impaired delivery of O_2_ to tissues, resulting in hypoxia, induced EPO production and therefore secondary erythrocytosis. Up to now, two types of erythrocytosis have been associated with mutations in hemoglobin genes, *HBA1*, *HBA2* (ECYT7) and *HBB* (ECYT6) respectively, however hemoglobin variants do have diverse pathophysiology [1,116]. Most of the Hb variants arise from amino acid substitutions that alter the α1β2 interface, the C- terminal end of the β chain or the 2,3-BPG binding site, which stabilize R state Hb or inhibit the affinity of Hb for its allosteric regulators that stimulate O_2_ release [117]. Very rare BPGM variants resulting in 2,3-BPG depletion and increased oxygen affinity of Hb have been described to cause ECYT8 [1]. A mild erythrocytosis has been reported also as a consequence of mutations in gene encoding cytochrome b5 reductase (B5R) enzyme [3,118], defects in *PKLR* gene, encoding glycolytic pyruvate kinase [11,119], and deficiency in the phosphofructokinase (PFK) enzyme [120]. Targeted exome NGS analysis of patients with idiopathic erythrocytosis by Camps et al. (2016) included four known disease-causing genes involved in Hb- O_2_ modulation (*BPGM*, *HBA1*, *HBA2*, *HBB*) and confirmed their role in the development of erythrocytosis [14]. Analysis of *PKLR* was included in the recent study by Kristan et al. (2021) as its role in the development of erythrocytosis has been previously indicated [11,121].

During our in silico research, we identified new candidate genes encoding hemoglobin modulators, membrane proteins and enzymes involved in glucose metabolism. Genes involved in the regulation of Hb mRNA expression were also investigated.

#### 2.3.1. Regulation at the RNA Level

GATA1 is a crucial regulator of not only genes encoding hemoglobin subunits, but also heme biosynthetic enzymes [122]. Transcription regulation of main hemoglobin genes *HBB*, *HBA1*, *HBA2*, and enzyme *BPGM* shall be revised further in detail.

#### 2.3.2. Regulation at the Protein Level

##### Hemoglobin Structure and Function Modulation

Erythrocytes of normal adults contain mainly adult hemoglobin (HbA), 2.5–3.5% HbA2, and  < 1% fetal hemoglobin (HbF). HbA is a heterotetramer composed of α and β globin subunits, each bound to a heme prosthetic group, iron (II) protoporphyrin IX [123]. Individual globin subunit forms a dimer with the unlike globin chain through two distinct interfaces, high-affinity α1β1 and low-affinity α1β2 [117]. Heme is a critical component of Hb which reversibly binds O_2_ through reduced state iron (Fe^2+^), located within a hydrophobic pocket on the external surface of the protein [124]. Oxidized (Fe^3+^) heme cannot bind O_2_ and is relatively unstable. It is reduced to functional (Fe^2+^) heme by the B5R enzyme [125]. α globin subunit is encoded by *HBA2* and *HBA1* paralogous genes [126], and the *HBB* gene encodes the β globin subunit. HbA2, consisting of two α and two δ globin subunits, has an unknown physiological role. However, the *HBD* gene that encodes the δ globin subunit, is a member of the *HBB* gene cluster [127]. HbF, consisting of two α and two γ globin subunits (genes *HBG1* and *HBG2*), is the dominant form expressed during fetal development with high oxygen affinity. HbF starts to decline just before birth and is gradually replaced by HbA over several months after birth. This switch is mediated by a replacement of *HBG* with *HBB* gene expression in definitive erythroid progenitors [128,129]. No erythrocytosis-causing variants of subunit coding genes *HBD* and *HBG* are known.

Oxygenation causes a change in the shape of globin chains. α1β2 interaction is destabilized by oxygen binding, resulting in a transition of the structure from the deoxygenated (T) state to the oxygenated (R) state, which facilitates the uptake of O_2_ by other subunits [117,130]. The O_2_ affinity properties of Hb are fine-tuned by several allosteric regulators, such as protons (H^+^), CO_2_, 2,3-diphosphoglycerate (2,3-BPG) and chloride ions (Cl^−^), which all promote the release of O_2_ [131]. CO_2_, taken up by erythrocytes, is metabolized by carbonic anhydrase enzyme (CA1) and, predominantly, by carbonic anhydrase II (CA2) to bicarbonate (HCO_3_^−^) and H^+^. CO_2_ combines with the N-terminal α-amino groups of HbA, while H^+^ binds histidine residues in HbA. This results in a conformational change of Hb, promoting O_2_ release. Besides, the efflux of HCO_3_^−^ from erythrocyte acidifies the cell and causes the release of oxygen from Hb. In the lungs these reactions are reversed, relatively low CO_2_ and thus high pH facilitates O_2_ binding to Hb [117,131]. 2,3-BPG is generated during glycolysis and its level is controlled by the bisphosphoglycerate mutase (BPGM) enzyme. 2,3- BPG binds in a central cavity of the Hb tetramer. Its binding converts the Hb to a low oxygen affinity state, allowing the efficient off-loading of O_2_ [1,117,130,132].

##### Gas Transport

Two membrane proteins are responsible for extremely high CO_2_ permeability of erythrocyte membrane, the water channel aquaporin-1 (AQP-1) and the Rhesus blood group type A glycoprotein (RhAG), both acting as CO_2_ channels [133]. Band 3 or anion exchanger 1, however, is the predominant glycoprotein of the erythrocyte membrane, responsible for HCO_3_^−^/Cl^−^ exchange, which is carried out by its membrane-spanning domain. This protein also constitutes a short C-terminal cytoplasmic domain, which binds CA2 enzyme, and a large N-terminal cytoplasmic domain, which binds glycolytic enzymes, Hb and hemichromes. Through its terminal domains, band 3 can maintain in close contact Hb and CA2 [131]. It has also been reported that the binding of deoxy-Hb to the cytoplasmic domain of band 3 causes the release of glycolytic enzymes from membrane to the cytosol to enhance glycolysis and 2,3-BPG production [134]. Besides, band 3 and Rh proteins have been demonstrated to form a band 3 macrocomplex [131]. Band 3 supposed to interact also with Glycophorin A (GPA), and it has been suggested that GPA promotes the cell surface expression and correct folding of band 3 for its proper function [135].

#### 2.3.3. Regulation at the Metabolic Level: Glycolysis and Oxygen Affinity Regulators

Erythrocytes during maturation loose nuclei, ribosomes and mitochondria, therefore they are completely dependent on glycolysis. Energy is generated via the anaerobic conversion of glucose to pyruvate or lactate by the Embden-Meyerhof pathway. The pathway is regulated by several glycolytic enzymes, including hexokinase (HK), glucose-6-phosphate isomerase (GPI), phosphofructokinase (PFK), aldolase, triosephosphate isomerase (TPI), glyceraldehyde-3-phosphate dehydrogenase (G-3-PD), phosphoglycerate kinase (PGK), pyruvate kinase (PK) and others [115]. Inside the Embden-Meyerhof pathway, erythrocytes possess a unique glycolytic bypass, the Rapoport-Luebbering shunt that avoids the PGK step of 1,3-diphosphoglycerate (1,3-BPG) conversion to 3-phosphoglycerate (3-PG). Instead, the bypass accounts for the synthesis and regulation of 2,3-BPG, the major glycolytic intermediate in erythrocytes, and its levels are about equal to the sum of the other glycolytic intermediates. The erythrocyte-specific BPGM is well known for catalyzing the formation of 2,3-BPG from 1,3-BPG and hydrolysis of 2,3-BPG to 3-PG, which then reenters the main glycolytic pathway [115,136]. Importantly, Cho et al. (2008) [136] identified a second enzyme component of the Rapoport–Luebbering shunt with 2,3-BPG phosphatase activity, multiple inositol polyphosphate phosphatase (MIPP1). Unlike BPGM, MIPP1 regulates 2,3-BPG levels by converting it to 2-PG and its capacity has been estimated to be equal to BPGM capacity.

As described above, many proteins regulate the final affinity of Hb for oxygen and its delivery to the tissues, and should be considered as causative factors, which can contribute to the development of erythrocytosis if their function is impaired. The importance of channels and transporters, allowing the translocation of Hb allosteric regulators to and out of erythrocytes, has to be emphasized. Besides, enzyme deficiencies distal to the 2,3-BPG step in the Embden-Meyerhof glycolytic pathway, such as HK, GPI, aldolase, and TPI, can also cause a decrease in 2,3-BPG levels and could thus contribute to erythrocytosis. The same applies to MIPP1, the second enzyme of Rapoport–Luebbering shunt with 2,3-BPG phosphatase activity, and PGK, an enzyme catalyzing the reversible conversion of 1,3-BPG to 3-PG [115,136]. Furthermore, erythrocyte metabolic adaptation to high-altitude hypoxia was recently reported to be regulated by an intracellular sphingolipid sphingosine 1-phosphate (S1P), a versatile bioactive intermediate metabolite of sphingosine. S1P is generated by sphingosine kinase 1 (SphK1) and SphK2, and irreversibly degraded by specific phosphatases or lyases. However, mature erythrocytes contain only Sphk1 but no S1P degrading enzymes, so they contain a much higher level of S1P. S1P has been shown to induce O_2_ release by direct binding to Hb, promoting anchoring of deoxygenated Hb to the membrane and consequently enhancing the release of membrane-bound glycolytic enzymes to the cytosol, inducing glycolysis and the production of 2,3-BPG [134]. Therefore, the contribution of this mechanism to the development of erythrocytosis should be further investigated.

## 3. Other Molecular Pathways Involved in Congenital Erythrocytosis

### 3.1. Regulation of Red Blood Cells Production and Degradation

Besides previously described regulation by oxygen sensing and EPO signaling, erythropoiesis is tightly controlled also by several other hormones and pathways directly or indirectly involved in erythrocyte maturation and degradation [124,125,137].

The potential new candidate that may contribute to the development of erythrocytosis are therefore hormones and their receptors that influence proliferation, survival and differentiation of erythrocytes, including macrophage colony-stimulating factor 1 (CSF1/CSF1R), Granulocyte-Macrophage Colony-Stimulating Factor 2 (CSF2/CSF2R) and interleukin 3 (IL3/IL3R). Several transcription factors in which dysregulation may also contribute to disease development are reviewed by Parisi et al. (2021) [138]. The Growth Factor Independent 1B Transcriptional Repressor (*GFI1B*) was confirmed as a novel candidate gene involved in the development of erythrocytosis in one of the previous studies [14]. Factors blocking the apoptosis of pre-erythrocytes should also be considered further, including regulation of autophagy, eryptosis and neocytolysis [139,140].

In addition to increased production of mature erythrocytes, reduced clearance of senescent erythrocytes may also contribute to the development of erythrocytosis [137].

### 3.2. Iron Metabolism

Being a component of heme, iron-sulfur proteins, and other enzymes, iron is required in many biological mechanisms, including oxygen transport, energy production, DNA synthesis, and cellular respiration [141]. Apart from EPO, iron is also a critical component for the production of mature erythrocytes. It is involved in hemoglobin synthesis, as well as it regulates the proliferation and differentiation of erythroblasts [142]. Most of the iron in the body is bound to Hb in erythrocytes, while blood plasma contains transferrin (TF) bound to iron, which is the exclusive source of iron for erythropoiesis in the bone marrow [143].

The stable concentration of circulating iron is maintained by dietary absorption, storage, and recycling from senescent red blood cells in macrophages [142]. Iron is stored in cytoplasmic ferritin (FT) in hepatocytes and macrophages of the liver and spleen and is readily mobilized when iron demand is increased [143]. Dietary absorption is tightly regulated by the small intestine wherein the iron uptake into the enterocyte depends on apical iron divalent metal transporter 1 (DMT1), and transition of iron to transferrin in the blood circulation depends on basolateral membrane exporter ferroportin (FPN) [143,144]. However, ferroportin and hormone hepcidin (HAMP) are two master regulators that maintain systemic iron metabolism and homeostasis. When hepcidin binds to ferroportin, this causes ferroportin degradation and consequently limits iron transport into the circulation. Hepcidin is predominately synthesized by the liver and abundantly produced in excess iron concentrations, while tissue hypoxia, iron deficiency, and ineffective erythropoiesis are the factors responsible for decreased transcription and production of hepcidin to enable ferroportin stabilization [141,142,143,145]. Hepcidin synthesis is regulated by SMAD signaling through several proteins and receptors, including transferrin receptor 1 (TFR1) and TFR2, and hemochromatosis protein (HFE) [141]. HFE forms a complex with TFR1 and TFR2. TRF2 functions as a sensor for the circulating transferrin-bound iron, and it is highly expressed in the liver. TFR1 is responsible for iron uptake from plasma transferrin to the cells and is ubiquitously expressed in most of the cells, especially in erythroid precursors [146,147]. However, in erythrocyte production, hepcidin synthesis is additionally regulated through erythroferrone hormone (ERFE) secreted by EPO-stimulated erythroblasts in JAK2/STAT5 dependent manner [141,147].

Being sensitive to cellular iron and oxygen levels, HIF2α has been reported as an essential factor to maintain systemic iron levels by regulating the transcription of genes encoding iron transporters, including DMT1 and FPN [144,148]. As demonstrated by Schwartz et al. (2019) [145], intestinal HIF2α also controls iron uptake during iron deficiency and hypoxia, and drives iron absorption during iron overload. Iron is a cofactor for PHD enzymes, and HIF2α stability and transcriptional response are regulated through hepcidin and ferroportin interaction that limits the activity of iron-dependent PHDs. Intestinal HIF2α signaling is activated during erythropoiesis and it has been shown to be critical for the increase in serum iron, necessary for efficient erythropoiesis [144].

The expression of transcripts of iron metabolism proteins that contain iron-responsive elements (IRE) sequences, such as TFR1, DMT1, FPN, and ferritin, is post-transcriptionally controlled by two cytosolic iron regulatory proteins, IRP1 and IRP2 [141,149]. In iron-deficient cells, IRPs bind with high affinity to IRE. HIF2α also contains an IRE sequence and it has been reported that the expression of HIF2α mRNA is exclusively regulated by IRP1, as IRP1-/- but not IRP2-/- mice developed erythrocytosis due to increased expression of EPO [149,150]. The involvement of TFR2 in erythropoiesis through its influence on EPOR signaling and terminal differentiation of erythroid progenitors has also been demonstrated. During the differentiation process, TFR2 and EPOR are co-expressed and TFR2 associates with EPOR in the endoplasmic reticulum for efficient transport of EPOR to the cell surface [146]. Furthermore, it has been proposed that HFE/TRF2 complex can potentiate MAPK signaling [143], and TFR1 can modulate EPO-dependent MAPK and PI3K signaling [73]. The connection of iron metabolic genes and erythrocytosis has also been demonstrated by a study by Biagetti et al. (2018) [12], where mutations in HFE were found in a relevant number of patients with idiopathic erythrocytosis. High incidence of HFE mutations in idiopathic erythrocytosis was indicated by several studies and the *HFE* gene together with other genes regulating iron metabolism (*HJV*, *HAMP*, *TFR2*, *SLC40A1*, *FTH1*, *TF*, *B2 M*, *CP*, *FTL*, *CDAN1*, *SEC23B*, *SLC25A38*, *STEAP3*, and *ALAS2*) were included in the genetic testing panel in our recent study [121,151]. Furthermore, decreased expression of hepcidin and increased expression of TFR1 and ERFE were observed in a transgenic mouse model with a JAK2 exon 12 mutation, frequently identified in patients with PV. High ERFE and low hepcidin together with elevated TFR1 enhance iron delivery to the sites of hematopoiesis to allow maximal production of red cells [152].

### 3.3. Other Mechanisms

It is well documented that testosterone therapy induces erythrocytosis [2,153]. In males normal values of hemoglobin are naturally higher, which confirms the assumption. Testosterone goes directly into the cell where it transforms into DHT (Dihydrotestosterone), with the help of 5-α reductase. DHT binds to the androgen receptor and together they are translocated into the cell nucleus, where dimer AR activates transcription of genes under the control of AR. The effect of other hormones, such as estrogen, growth hormone, thyroid hormones should also be addressed.

Genes involved in the metabolism of metals cobalt, nickel and manganese (*SLC30A10*), calcium (*PIEZO1*) and chloride ions must be carefully revised. Transcription of the erythropoietin may be modulated by metals (cobalt and nickel), which can substitute iron in the porphyrin ring [154].

Environmental manganese (Mn) is toxic to the body, therefore manganese transporter encoded by Solute Carrier Family 30 Member 10 *(SLC30A10)* is crucial to maintain an appropriate manganese level in the body. Mutations in *SLC30A10* were associated with Syndrome of Hepatic Cirrhosis, Dystonia, Polycythemia and Hypermanganesemia [155]. Therefore, the level of Mn and mutations of *SLC30A10* should be revised in erythrocytosis patients.

Several recent publications have suggested the role of protein Piezo Type Mechanosensitive Ion Channel Component 1 (PIEZO1), the nonselective cation channel located on the membrane of the erythrocytes, which is activated by mechanic pressure. The gain-of-function mutation of gene encoding this channel was found in some cases of idiopathic erythrocytosis and xerocytosis [156,157,158]. Individuals with hereditary xerocytosis develop age-onset iron overload, linking the PIEZO1 to iron metabolism [159]. Another study suggests the role of PIEZO1 in the response of erythrocytes to mechanical stress and indicates its role also in early erythropoiesis [160].

## 4. Conclusions

Erythrocytosis is a rare hematological disorder that arises from an imbalance in homeostatic mechanisms of erythropoiesis and oxygen homeostasis. The main regulatory pathways in the process of oxygen homeostasis are still poorly understood, but this knowledge is necessary to identify new potential causative factors and to improve the current diagnostic methods. This review aimed to discover additional molecular mechanisms correlated with erythropoiesis and therefore erythrocytosis. We identified genes involved in post-translational modifications, nuclear and membrane transport, transcriptional regulation, positive and negative regulatory mechanisms of signal transduction, and erythrocyte glucose metabolism, which can modulate the stability or activity of key erythropoiesis genes, including *EPAS1*, *EPO*, *EPOR*, *HBA* and *HBB*, and their partners, and consequently influence the process of erythrocyte production. Besides, several studies have demonstrated the interplay between disrupted iron homeostasis and erythrocytosis. Since iron is a critical component in the process of erythrocyte production, this represents another possible connection to erythrocytosis of unknown cause that should be further investigated. Epigenetics may also contribute to the development of disease. It must be taken into account, that besides monogenic, digenic or polygenic origin of the disorder shall also be addressed. Targets identified within this in silico review however need to be identified and confirmed on clinical samples and tested by functional analysis.

The diagnostic workflow should begin with a detailed review of clinical data and exclusion of any acquired cases for erythrocytosis; such as compensatory erythrocytosis due to chronic disease or polycythemia vera due to a *JAK2* mutation. Only patients who remain idiopathic should undergo targeted NGS genetic testing covering all genes previously associated with erythrocytosis. In addition to sequencing exome region, attention should be paid to noncoding regions, including 5’ and 3’ UTRs, promoters, enhancers, introns and miRNA binding sites. If no disease driving genetic variant is identified, families with multiple members with erythrocytosis should be referred for whole-exome sequencing (WES) or whole-genome sequencing (WGS). Newly identified variants should be confirmed as causative by functional analysis. Attention shall be addressed to cumulative effect of several variants.

The limit of the current studies is, that the role of several molecules is studied in detail in cancer, while the information on the normal tissue is not available. Therefore, their enrolment in the regulatory mechanisms in healthy tissue must be considered with caution. However, mathematical modeling will expedite future research, as a model of all pathways will facilitate to understand the roles of specific genes/gene networks in the development of erythrocytosis.

## Figures and Tables

**Figure 1 genes-12-01150-f001:**
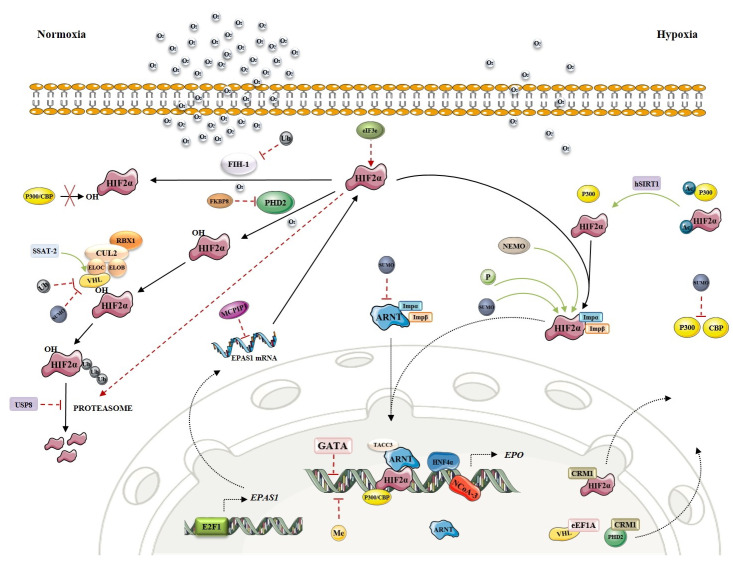
Molecular mechanisms of HIF-EPO pathway. The red line indicates an inhibitory effect. Green line indicates the activation/stabilization effect. Black line indicates transition. Circles denote post-translational mechanisms: P—Phosphorylation, Ac—Acetylation, Me—Methylation, SUMO—Sumoylation, Ub—Ubiquitynation (Figure was created using SMART servier medical art [18]).

**Figure 2 genes-12-01150-f002:**
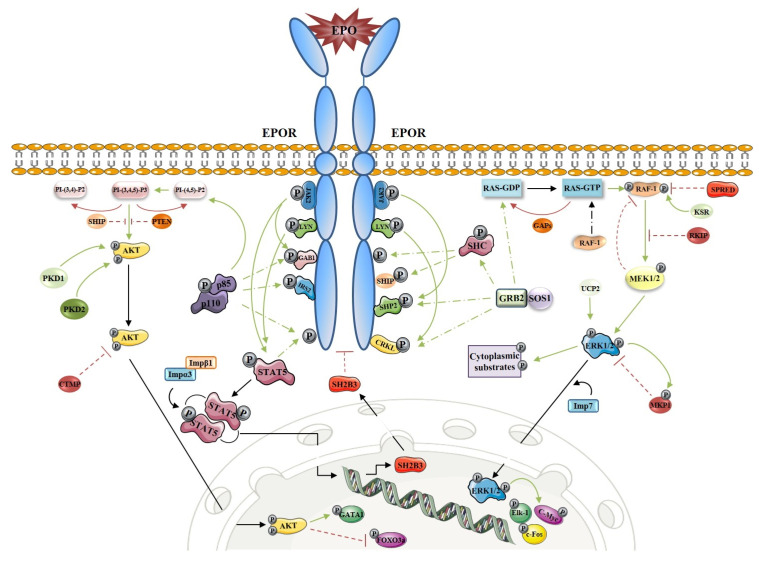
Molecular mechanisms of EPO-EPOR signal transduction pathways: STAT5 pathway, PI3K pathway and MAPK pathway. Red line indicates an inhibitory effect. Green line indicates the activation effect. Black line indicates transition (Figure was created using SMART servier medical art [18]. Figure adapted from [61,62]).

**Figure 3 genes-12-01150-f003:**
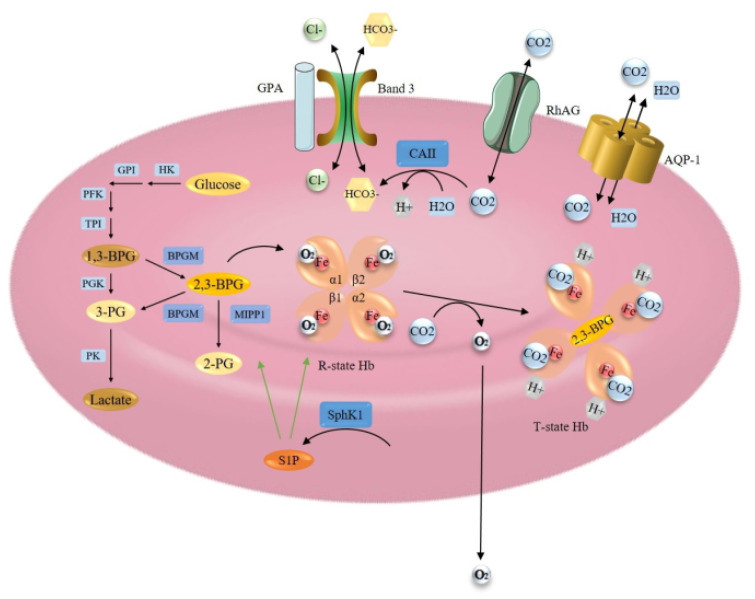
Erythrocyte and hemoglobin structure and function modulation, gas transport and glycolysis pathway, including Rapoport-Luebering shunt. Green line indicates the activation effect. Black line indicates transition (Figure was created using Reactome database [4]).

## Data Availability

Not applicable.

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
