# Peer review of "Molecular Pathways Involved in the Development of Congenital Erythrocytosis"

_genes, 2021, doi:10.3390/genes12081150_

Round 1
Reviewer 1 Report
This is an outstanding review of the current knowledge re causes of congenital erythrocytosis. Very easy to read and well referenced. The closing sentences on line 552 states -“Genes involved in the metabolism of metals cobalt, nickel and manganese 552 (SLC30A10), calcium (PIEZO1) and chloride ions must carefully be revised [143].” The paper by Bunn et al does not discuss a role for calcium- perhaps there is another reference. Although the exact mechanism is unknown two recent publications suggest role of calcium channel protein PIEZO1 in some cases of idiopathic erythrocytosis ( Knight T et al - Pediatr Hematol Oncol. 2019 Aug;36(5):317-326. doi: 10.1080/08880018.2019.1637984. Epub 2019 Jul 12.PMID: 31298594; Filser M et al - Blood. 2021 Apr 1;137(13):1828-1832. doi: 10.1182/blood.2020008424)
Other comments:
In figure 3 suggest lightening the background color intensity of the red cells- it is hard to see the genes and pathways. Also suggest adding PIEZO1 as membrane protein.
Author Response
GENERAL:
We thank all reviewers for detailed revision of the paper and indicated corrections, that improved our paper. All corrections suggested by reviewers including additional minor corrections by authors are marked with traced changes. Proposed new references were included and reference list updated accordingly.
REVIEWER 1
Comments and Suggestions for Authors
This is an outstanding review of the current knowledge re causes of congenital erythrocytosis. Very easy to read and well referenced. The closing sentences on line 552 states -“Genes involved in the metabolism of metals cobalt, nickel and manganese 552 (SLC30A10), calcium (PIEZO1) and chloride ions must carefully be revised [143].” The paper by Bunn et al does not discuss a role for calcium- perhaps there is another reference.
The statement regarding reference was corrected.
Although the exact mechanism is unknown two recent publications suggest role of calcium channel protein PIEZO1 in some cases of idiopathic erythrocytosis (Knight T et al - Pediatr Hematol Oncol. 2019 Aug;36(5):317-326. doi: 10.1080/08880018.2019.1637984. Epub 2019 Jul 12.PMID: 31298594; Filser M et al - Blood. 2021 Apr 1;137(13):1828-1832. doi: 10.1182/blood.2020008424)
The description was elaborated and references included (p. 14, line 1193-1199):
“Several recent publications suggest role of protein Piezo Type Mechanosensitive Ion Channel Component 1 (PIEZO1), the nonselective cation channel located on the mem-brane of the erythrocytes activated by mechanic pressure. The gain-of-function mutation of this channel was found in some cases of idiopathic erythrocytosis and xerocytosis [155-157]. Individuals with hereditary xerocytosis develop age-onset iron overload, link-ing the PIEZO1 to iron metabolism [158]. Another study suggested the role of PIEZO1 in response of erythrocytes to mechanical stress and indicates its role also in early erythropoiesis [159].”
- Knight T et al Mild erythrocytosis as a presenting manifestation of PIEZO1 associated erythrocyte volume disorders Pediatr Hematol Oncol. 2019 Aug;36(5):317-326. doi: 10.1080/08880018.2019.1637984.
- Filser M et al - Increased incidence of germline PIEZO1 mutations in individuals with idiopathic erythrocytosis Blood. 2021 Apr 1;137(13):1828-1832. doi: 10.1182/blood.2020008424)
- Kiger, Lydie Oliveira, Corinne Guitton, Laurence Bendélac, Kaldoun Ghazal, Valérie Proulle, Frédéric Galacteros, Christophe Junot, François Fenaille, Paul-Henri Roméo, Loic Garçon, Véronique Picard. Piezo1-xerocytosis red cell metabolome shows impaired glycolysis and increased hemoglobin oxygen affinity. Laurent. Blood Adv (2021) 5 (1): 84–88.
- Ma S, Dubin AE, Zhang Y, Mousavi SAR, Wang Y, Coombs AM, Loud M, Andolfo I, Patapoutian A. A role of PIEZO1 in iron metabolism in mice and humans. Cell. 2021 Feb 18;184(4):969-982.e13. doi: 10.1016/j.cell.2021.01.024.
- Jankovsky N, Caulier A, Demagny J, Guitton C, Djordjevic S, Lebon D, Ouled-Haddou H, Picard V, Garçon L. Recent advances in the pathophysiology of PIEZO1-related hereditary xerocytosis. Am J Hematol. 2021 Aug 1;96(8):1017-1026. doi: 10.1002/ajh.26192.
Other comments:
In figure 3 suggest lightening the background color intensity of the red cells- it is hard to see the genes and pathways. Also suggest adding PIEZO1 as membrane protein.
Figure 3 was replaced and corrected as indicated by reviewer.
Reviewer 2 Report
According to my opinion, the title of the review neither fully corresponds with the content nor with aim of the review. The authors are not presenting novel molecular pathways involved in the development of congenital erythrocytoses. They only provide an overview of different pathways and/or levels of the regulation of erythropoiesis. In most of the cases, no direct evidence for the involvement of proposed novel molecular targets as causative genes in erythrocytosis is provided. It is generally recognized that novel potential erythrocytosis causing genes will be associated with the processes responsible for the regulation of erythropoiesis, but without functional studies and/or identification of gene mutations in erythrocytosis patients it is all just a hypothesis. They neither suggest a work-flow for testing the patients for those potentially novel candidates and/or for improvement the diagnostics of idiopathic erythrocytosis. On the other hand, they do not explicitly state that rare cases of erythrocytosis associated with mutations in OP9 or weakly activating JAK2 germline mutations have been described. They neither present an important recent finding that some Piezo1- hereditary xerocytosis patients present with erythrocytosis and show impaired metabolome and glycolysis. Finally, the authors recently published a similar article on genes and pathways involved in erythrocytosis development (published in Blood Transfusion, 2020) and that paper better covers the topic than this one submitted to Genes.
Author Response
GENERAL:
We thank all reviewers for detailed revision of the paper and indicated corrections, that improved our paper. All corrections suggested by reviewers including additional minor corrections by authors are marked with traced changes. Proposed new references were included and reference list updated accordingly.
REVIWER 2
Comments and Suggestions for Authors
According to my opinion, the title of the review neither fully corresponds with the content nor with aim of the review. The authors are not presenting novel molecular pathways involved in the development of congenital erythrocytoses. They only provide an overview of different pathways and/or levels of the regulation of erythropoiesis.
The title of the paper and chapters were corrected in aim to better represent the topic covered within this review article. New title of the paper is:
»Molecular pathways involved in the development of congenital erythrocytosis«
In most of the cases, no direct evidence for the involvement of proposed novel molecular targets as causative genes in erythrocytosis is provided. It is generally recognized that novel potential erythrocytosis causing genes will be associated with the processes responsible for the regulation of erythropoiesis, but without functional studies and/or identification of gene mutations in erythrocytosis patients it is all just a hypothesis.
This is a review where in silico methods were used in aim to review all currently know molecular pathways. Newly discovered genes need to be later confirmed by in vitro mutagenesis and functional studies.
They neither suggest a work-flow for testing the patients for those potentially novel candidates and/or for improvement the diagnostics of idiopathic erythrocytosis.
Diagnostic workflow is proposed in conclusion (p 14, line 1074-1083).
“The diagnostic workflow should begin with a detailed review of clinical data and exclusion of any acquired cases for erythrocytosis; such as compensatory erythrocytosis due to cronic disease or policitemia vera due to a JAK2 mutation. Only patients who remain idiopathic should undergo extensive NGS genetic testing all genes previously associated with erythrocytosis. In addition to exome region sequencing, attention should be paid to noncoding regions, including 5' and 3' UTRs, promoters, enhancers, and introns. If no genetic cause is identified, families with multiple members with erythrocytosis should be referred for whole-exome sequencing (WES) or whole-genome sequencing (WGS). Newly identified variants should be confirmed as causative by functional analysis. Attention shall be addressed to cumulative effect of several variants.”
On the other hand, they do not explicitly state that rare cases of erythrocytosis associated with mutations in OP9 or weakly activating JAK2 germline mutations have been described.
The description was modified and statement of OS9 and BHLHE41 derived erythrocytosis is added and new references included (p. 2, line 84-94):
“Targeted exome NGS analysis of patients with idiopathic erythrocytosis by Camps et al. (2016), including known disease-causing oxygen-sensing genes (EPAS1, EGLN1, VHL) and novel candidate genes from oxygen-sensing pathway (EPO, HIF1A, HIF3A, HIF1AN, EGLN2, EGLN3, BHLHE41, OS9, ZNF197, KDM6A) confirmed variants in all known dis-ease-causing genes and in five novel candidate genes. Novel candidate gene associated with ECYT identified within this study were EPO, EPAS1 paralogues HIF3A, EGLN1 pa-ralogue EGLN2, and two factors associated with HIF; OS9 Endoplasmic Reticulum Lectin (OS9) and class E basic helix-loop-helix protein 41 (BHLHE41 or SHARP1) [14]. Further-more, the role of DNA non-coding regions in development of erythrocytosis should further be considered, as their role has already been confirmed in EPO and VHL by functional studies [15, 16].”
- Zmajkovic J, Lundberg P, Nienhold R, Torgersen ML, Sundan A, Waage A, Skoda RC.N. A Gain-of-Function Mutation in EPO in Familial Erythrocytosis. Engl J Med. 2018 Mar 8;378(10):924-930. doi: 10.1056/NEJMoa1709064.
- Lenglet M, Robriquet F, Schwarz K, Camps C, Couturier A, Hoogewijs D, Buffet A, Knight SJL, Gad S, Couvé S, Chesnel F, Pacault M, Lindenbaum P, Job S, Dumont S, Besnard T, Cornec M, Dreau H, Pentony M, Kvikstad E, Deveaux S, Burnichon N, Ferlicot S, Vilaine M, Mazzella JM, Airaud F, Garrec C, Heidet L, Irtan S, Mantadakis E, Bouchireb K, Debatin KM, Redon R, Bezieau S, Bressac-de Paillerets B, Teh BT, Girodon F, Randi ML, Putti MC, Bours V, Van Wijk R, Göthert JR, Kattamis A, Janin N, Bento C, Taylor JC, Arlot-Bonnemains Y, Richard S, Gimenez-Roqueplo AP, Cario H, Gardie B. Identification of a new VHL exon and complex splicing alterations in familial erythrocytosis or von Hippel-Lindau disease. Blood. 2018 Aug 2;132(5):469-483. doi: 10.1182/blood-2018-03-838235.
New citations on JAK2 germline and other mutations were included in the paper (p. 6, line 413-416):
- Kapralova K, Horvathova M, Pecquet C, Fialova Kucerova J, Pospisilova D, Leroy E, Kralova B, Milosevic Feenstra JD, Schischlik F, Kralovics R, Constantinescu SN, Divoky V. Cooperation of germ line JAK2 mutations E846D and R1063H in hereditary erythrocytosis with megakaryocytic atypia. Blood. 2016 Sep 8;128(10):1418-23. doi: 10.1182/blood-2016-02-698951. Epub 2016 Jul 7.
- Ruth Stuckey, María Teresa Gómez-Casares. Recent Advances in the Use of Molecular Analyses to Inform the Diagnosis and Prognosis of Patients with Polycythaemia Vera. Int J Mol Sci 2021 May 10;22(9):5042. doi: 10.3390/ijms22095042.
They neither present an important recent finding that some Piezo1- hereditary xerocytosis patients present with erythrocytosis and show impaired metabolome and glycolysis.
The description was elaborated and proposed references included (p. 14, line 1193-1199):
“Several recent publications suggest role of protein Piezo Type Mechanosensitive Ion Channel Component 1 (PIEZO1), the nonselective cation channel located on the mem-brane of the erythrocytes activated by mechanic pressure. The gain-of-function mutation of this channel was found in some cases of idiopathic erythrocytosis and xerocytosis [155-157]. Individuals with hereditary xerocytosis develop age-onset iron overload, link-ing the PIEZO1 to iron metabolism [158]. Another study suggested the role of PIEZO1 in response of erythrocytes to mechanical stress and indicates its role also in early erythropoiesis [159].”
- Knight T et al Mild erythrocytosis as a presenting manifestation of PIEZO1 associated erythrocyte volume disorders Pediatr Hematol Oncol. 2019 Aug;36(5):317-326. doi: 10.1080/08880018.2019.1637984.
- Filser M et al - Increased incidence of germline PIEZO1 mutations in individuals with idiopathic erythrocytosis Blood. 2021 Apr 1;137(13):1828-1832. doi: 10.1182/blood.2020008424)
- Kiger, Lydie Oliveira, Corinne Guitton, Laurence Bendélac, Kaldoun Ghazal, Valérie Proulle, Frédéric Galacteros, Christophe Junot, François Fenaille, Paul-Henri Roméo, Loic Garçon, Véronique Picard. Piezo1-xerocytosis red cell metabolome shows impaired glycolysis and increased hemoglobin oxygen affinity. Laurent. Blood Adv (2021) 5 (1): 84–88.
- Ma S, Dubin AE, Zhang Y, Mousavi SAR, Wang Y, Coombs AM, Loud M, Andolfo I, Patapoutian A. A role of PIEZO1 in iron metabolism in mice and humans. Cell. 2021 Feb 18;184(4):969-982.e13. doi: 10.1016/j.cell.2021.01.024.
- Jankovsky N, Caulier A, Demagny J, Guitton C, Djordjevic S, Lebon D, Ouled-Haddou H, Picard V, Garçon L. Recent advances in the pathophysiology of PIEZO1-related hereditary xerocytosis. Am J Hematol. 2021 Aug 1;96(8):1017-1026. doi: 10.1002/ajh.26192.
Finally, the authors recently published a similar article on genes and pathways involved in erythrocytosis development (published in Blood Transfusion, 2020) and that paper better covers the topic than this one submitted to Genes.
Paper in Blood Transfusion is an overall revision of genetic background of know erythrocytosis-causing genes. While in current paper we systematically explore additional pathways involved in regulation of know erythrocytosis-causing genes/proteins and their partners.
Reviewer 3 Report
The authors of the review untitled « Known and novel molecular pathways involved in the development of congenital erythrocytosis » described all the biological pathways that regulate the production of red blood cells and may be potential actors in the development of erythrocytosis.
This review is well written and rather complete. It opens up new directions for consideration of different targets that could be explored in the diagnosis of congenital erythrocytosis.
Below some minor modifications requested:
- Some pathways are listed as potential new candidates in the occurrence of erythrocytosis. However, some clinical data (or reviews) already support these hypotheses. These articles should be cited:
- SH2B3:
- McMullin MF, Cario H. LNK mutations and myeloproliferative disorders. Am J Hematol. 2016 Feb;91(2):248-51.
- HFE:
- Giacomo Biagetti, Mark Catherwood , Nuala Robson, Irene Bertozzi, Elisabetta Cosi, Mary F McMullin, Maria L Randi. HFE mutations in idiopathic erythrocytosis. Br J Haematol. 2018 Apr;181(2):270-272.
- Burlet B, Bourgeois V, Buriller C, Aral B, Airaud F, Gardie B, Girodon High HFE mutation incidence in idiopathic erythrocytosis. Br J Haematol. 2018 Nov 8.
- PIEZO1:
- Filser M, Giansily-Blaizot M, Grenier M, Monedero Alonso D, Bouyer G, Laurent Pérès, Egée S, Aral B, Airaud F, Da Costa L, Picard V, Cougoul P, Palach M, Béziau S, Garrec C, Patricia Aguilar-Martinez P, Gardie B*, Girodon F*. Increased incidence of germline PIEZO1 mutations in individuals with idiopathic erythrocytosis. Letter. Blood, 2021 Apr 1;137(13):1828-1832.
- Line 62, please refer to the corresponding article.
- Lines 109, 119, 120394, 397, 399, please put the name of the genes in italics.
- SH2B3:
Author Response
GENERAL:
We thank all reviewers for detailed revision of the paper and indicated corrections, that improved our paper. All corrections suggested by reviewers including additional minor corrections by authors are marked with traced changes. Proposed new references were included and reference list updated accordingly.
REVIWER 3
Comments and Suggestions for Authors
The authors of the review untitled « Known and novel molecular pathways involved in the development of congenital erythrocytosis » described all the biological pathways that regulate the production of red blood cells and may be potential actors in the development of erythrocytosis.
This review is well written and rather complete. It opens up new directions for consideration of different targets that could be explored in the diagnosis of congenital erythrocytosis.
Below some minor modifications requested:
- Some pathways are listed as potential new candidates in the occurrence of erythrocytosis. However, some clinical data (or reviews) already support these hypotheses. These articles should be cited:
- SH2B3:
McMullin MF, Cario H. LNK mutations and myeloproliferative disorders. Am J Hematol. 2016 Feb;91(2):248-51.
The proposed citation was included in the paper.
- HFE:
Giacomo Biagetti, Mark Catherwood , Nuala Robson, Irene Bertozzi, Elisabetta Cosi, Mary F McMullin, Maria L Randi. HFE mutations in idiopathic erythrocytosis. Br J Haematol. 2018 Apr;181(2):270-272.
Burlet B, Bourgeois V, Buriller C, Aral B, Airaud F, Gardie B, Girodon High HFE mutation incidence in idiopathic erythrocytosis. Br J Haematol. 2018 Nov 8.
The proposed reference is already included in the paper, listed as ref. 140, now 12. New reference was also included (p. 13, line 1058-1062):
“High incidence of HFE mutations in idiopathic erythrocytosis was indicated by several studies and HFE gene together with other genes regulating iron metabolism (HJV, HAMP, TFR2, SLC40A1, FTH1, TF, B2 M, CP, FTL, CDAN1, SEC23B, SLC25A38, STEAP3 and ALAS2) were included in genetic testing panel in our recent study [151, 121].
- Burlet B, Bourgeois V, Buriller C, Aral B, Airaud F, Gardie B, Girodon High HFE mutation incidence in idiopathic erythrocytosis. Br J Haematol. 2018.”
- PIEZO1:
Filser M, Giansily-Blaizot M, Grenier M, Monedero Alonso D, Bouyer G, Laurent Pérès, Egée S, Aral B, Airaud F, Da Costa L, Picard V, Cougoul P, Palach M, Béziau S, Garrec C, Patricia Aguilar-Martinez P, Gardie B*, Girodon F*. Increased incidence of germline PIEZO1 mutations in individuals with idiopathic erythrocytosis. Letter. Blood, 2021 Apr 1;137(13):1828-1832.
The description was elaborated and proposed references included (p. 14, line 1193-1199):
“Several recent publications suggest role of protein Piezo Type Mechanosensitive Ion Channel Component 1 (PIEZO1), the nonselective cation channel located on the mem-brane of the erythrocytes activated by mechanic pressure. The gain-of-function mutation of this channel was found in some cases of idiopathic erythrocytosis and xerocytosis [155-157]. Individuals with hereditary xerocytosis develop age-onset iron overload, link-ing the PIEZO1 to iron metabolism [158]. Another study suggested the role of PIEZO1 in response of erythrocytes to mechanical stress and indicates its role also in early erythropoiesis [159].”
- Knight T et al Mild erythrocytosis as a presenting manifestation of PIEZO1 associated erythrocyte volume disorders Pediatr Hematol Oncol. 2019 Aug;36(5):317-326. doi: 10.1080/08880018.2019.1637984.
- Filser M et al - Increased incidence of germline PIEZO1 mutations in individuals with idiopathic erythrocytosis Blood. 2021 Apr 1;137(13):1828-1832. doi: 10.1182/blood.2020008424)
- Kiger, Lydie Oliveira, Corinne Guitton, Laurence Bendélac, Kaldoun Ghazal, Valérie Proulle, Frédéric Galacteros, Christophe Junot, François Fenaille, Paul-Henri Roméo, Loic Garçon, Véronique Picard. Piezo1-xerocytosis red cell metabolome shows impaired glycolysis and increased hemoglobin oxygen affinity. Laurent. Blood Adv (2021) 5 (1): 84–88.
- Ma S, Dubin AE, Zhang Y, Mousavi SAR, Wang Y, Coombs AM, Loud M, Andolfo I, Patapoutian A. A role of PIEZO1 in iron metabolism in mice and humans. Cell. 2021 Feb 18;184(4):969-982.e13. doi: 10.1016/j.cell.2021.01.024.
- Jankovsky N, Caulier A, Demagny J, Guitton C, Djordjevic S, Lebon D, Ouled-Haddou H, Picard V, Garçon L. Recent advances in the pathophysiology of PIEZO1-related hereditary xerocytosis. Am J Hematol. 2021 Aug 1;96(8):1017-1026. doi: 10.1002/ajh.26192.
- Line 62, please refer to the corresponding article.
References added, we included references 11 and 12, previously stated as 108 and 140 (p. 2, line 65):
- Beutler, E.; Westwood, B.; Van Zwieten, R.; Roos, D. G→t Transition at Cdna Nt 110 (K37q) in the Pklr (Pyruvate Ki-nase) Gene Is the Molecular Basis of a Case of Hereditary Increase of Red Blood Cell Atp. Hum. Mutat. 1997, 9 (3), 282–285.
- Biagetti, G.; Catherwood, M.; Robson, N.; Bertozzi, I.; Cosi, E.; McMullin, M. F.; Randi, M. L. HFE Mutations in Idio-pathic Erythrocytosis. Br. J. Haematol. 2018, 181 (2), 270–272. https://doi.org/10.1111/bjh.14555.
- Lines 109, 119, 120394, 397, 399, please put the name of the genes in italics.
Corrected.
Reviewer 4 Report
Jana Tomc et al review possible genetic causes for the occurrence of congenital erythrocytosis.
Based on a comprehensive review of the literature and different in silico tools, they suggest molecular pathways that could be involved in congenital erythrocytosis.
The article is well written and the bibliographical references are generally appropriate. The figures are comprehensive and informative.
Minor points:
- While in silico studies are of theoretical interest to consider the various genetic targets, there is currently very little (if any) experimental or patient evidence to support the authors' claims, which remain highly speculative.
In the last paragraph, the authors should develop, for manganese overload, the mutations of the SLC30A10 gene (Tuschl et coll The American Journal of Human Genetics90, 457–466, March 9, 2012), since there is an abundance of literature on the subject.
In the same paragraph, the authors should also develop the mechanisms associated with the PIEZO1 mutations that have been recently reported (Kiger L and coll Blood Adv 2021)
Author Response
GENERAL:
We thank all reviewers for detailed revision of the paper and indicated corrections, that improved our paper. All corrections suggested by reviewers including additional minor corrections by authors are marked with traced changes. Proposed new references were included and reference list updated accordingly.
REVIWER 4
Comments and Suggestions for Authors
Jana Tomc et al review possible genetic causes for the occurrence of congenital erythrocytosis.
Based on a comprehensive review of the literature and different in silico tools, they suggest molecular pathways that could be involved in congenital erythrocytosis.
The article is well written and the bibliographical references are generally appropriate. The figures are comprehensive and informative.
Minor points:
- While in silico studies are of theoretical interest to consider the various genetic targets, there is currently very little (if any) experimental or patient evidence to support the authors' claims, which remain highly speculative.
In this review in silico methods were used in aim to review all currently know molecular pathways. Newly discovered genes need to be confirmed by in vitro mutagenesis and functional studies.
In the last paragraph, the authors should develop, for manganese overload, the mutations of the SLC30A10 gene (Tuschl et coll The American Journal of Human Genetics90, 457–466, March 9, 2012), since there is an abundance of literature on the subject.
The description was elaborated and proposed reference is included (p. 13, line 1076-1079):
“Environmental manganese (Mn) is toxic to the body, therefore manganese transporter encoded by Solute Carrier Family 30 Member 10 (SLC30A10) is crucial to maintains ap-propriate manganese levels in the body. Mutations in SLC30A10 were associated with Syndrome of Hepatic Cirrhosis, Dystonia, Polycythemia and Hypermanganesemia [154]. Therefore, the level of Mn and mutations of SLC30A10 should be revised in erythrocytosis patients.”
In the same paragraph, the authors should also develop the mechanisms associated with the PIEZO1 mutations that have been recently reported (Kiger L and coll Blood Adv 2021)
The description was elaborated and proposed references included (p. 14, line 1193-1199):
“Several recent publications suggest role of protein Piezo Type Mechanosensitive Ion Channel Component 1 (PIEZO1), the nonselective cation channel located on the mem-brane of the erythrocytes activated by mechanic pressure. The gain-of-function mutation of this channel was found in some cases of idiopathic erythrocytosis and xerocytosis [155-157]. Individuals with hereditary xerocytosis develop age-onset iron overload, link-ing the PIEZO1 to iron metabolism [158]. Another study suggested the role of PIEZO1 in response of erythrocytes to mechanical stress and indicates its role also in early erythropoiesis [159].”
- Knight T et al Mild erythrocytosis as a presenting manifestation of PIEZO1 associated erythrocyte volume disorders Pediatr Hematol Oncol. 2019 Aug;36(5):317-326. doi: 10.1080/08880018.2019.1637984.
- Filser M et al - Increased incidence of germline PIEZO1 mutations in individuals with idiopathic erythrocytosis Blood. 2021 Apr 1;137(13):1828-1832. doi: 10.1182/blood.2020008424)
- Kiger, Lydie Oliveira, Corinne Guitton, Laurence Bendélac, Kaldoun Ghazal, Valérie Proulle, Frédéric Galacteros, Christophe Junot, François Fenaille, Paul-Henri Roméo, Loic Garçon, Véronique Picard. Piezo1-xerocytosis red cell metabolome shows impaired glycolysis and increased hemoglobin oxygen affinity. Laurent. Blood Adv (2021) 5 (1): 84–88.
- Ma S, Dubin AE, Zhang Y, Mousavi SAR, Wang Y, Coombs AM, Loud M, Andolfo I, Patapoutian A. A role of PIEZO1 in iron metabolism in mice and humans. Cell. 2021 Feb 18;184(4):969-982.e13. doi: 10.1016/j.cell.2021.01.024.
- Jankovsky N, Caulier A, Demagny J, Guitton C, Djordjevic S, Lebon D, Ouled-Haddou H, Picard V, Garçon L. Recent advances in the pathophysiology of PIEZO1-related hereditary xerocytosis. Am J Hematol. 2021 Aug 1;96(8):1017-1026. doi: 10.1002/ajh.26192.